# Abundance, Size Structure, and Growth of the Invasive Blue Crab *Callinectes sapidus* in the Lesina Lagoon, Southern Adriatic Sea

**DOI:** 10.3390/biology13121051

**Published:** 2024-12-15

**Authors:** Giorgio Mancinelli, Nicola Lago, Tommaso Scirocco, Oscar Antonio Lillo, Raffaele De Giorgi, Lorenzo Doria, Emanuele Mancini, Francesco Mancini, Luigi Potenza, Lucrezia Cilenti

**Affiliations:** 1Department of Biological and Environmental Sciences and Technologies (DiSTeBA), University of Salento, 73100 Lecce, Italy; giorgio.mancinelli@unisalento.it (G.M.); raffaele.degiorgi1@unisalento.it (R.D.G.); lorenzo.doria@unisalento.it (L.D.); emanuele.mancini@unisalento.it (E.M.); luigi.potenza@unisalento.it (L.P.); 2National Biodiversity Future Center (NBFC), 90133 Palermo, Italy; 3Consorzio Nazionale Interuniversitario per le Scienze del Mare (CoNISMa), 00196 Roma, Italy; 4Istituto Zooprofilattico Sperimentale delle Venezie (IZSVe), 35020 Legnaro, Italy; nlago@izsvenezie.it; 5National Research Council (CNR), Institute for Biological Resources and Marine Biotechnologies (CNR-IRBIM), 71010 Lesina, Italy; tommaso.scirocco@cnr.it; 6Metaponto Research Centre, Regional Agency for the Environmental Prevention and Protection of Basilicata (ARPAB), 75012 Metaponto, Italy; antoniooscar.lillo@farbas.it; 7Environmental Research Foundation of Basilicata Region (FARBAS), 85100 Potenza, Italy; 8International Centre for Advanced Mediterranean Agronomic Studies (CIHEAM-Bari), 70010 Valenzano, Italy; mancini@iamb.it; 9National Research Council (CNR), Institute of Sciences of Food Production (ISPA), 71121 Foggia, Italy

**Keywords:** invasive species, fishery biology, CPUE, sex ratio, size structure, morphological maturity, growth dynamics, Mediterranean Sea

## Abstract

A comprehensive assessment of the life history parameters of the invasive Atlantic blue crab *Callinectes sapidus* in the Lesina Lagoon (Adriatic Sea, SE Italy) in terms of abundance, sex ratio, size–frequency distribution, morphological maturation, growth, and mortality was provided adopting a suite of analytical techniques. The results of the present study can provide a valuable knowledge basis for future comparative investigations on the fishery biology of *C. sapidus* in Mediterranean waters.

## 1. Introduction

The Atlantic blue crab *Callinectes sapidus* Rathbun, 1896 (“blue crab” hereafter) is a portunid brachyuran characterized among other members of the genus by a trans-hemispheric distribution along the western coasts of the Atlantic Ocean, ranging from the Gulf of Maine through the Gulf of Mexico to Venezuela, Brazil, and Argentina [1,2,3]. In its native range the blue crab is is recognized as a keystone species regulating trophic cascades and the transfer of energy between the benthic and pelagic compartments [4,5] and represents the object of a major commercial and recreational fishery [6,7,8]. Hence, a huge body of information exists on the biology and ecology of the species primarily from the USA—among the thousands of publications available to date, the book “the Blue Crab: *Callinectes sapidus*” [9] represents an outstanding example and provides a comprehensive research reference—and from central and south American countries [8,10,11,12,13,14].

In the past century, unsuccessful attempts have been made to deliberately introduce the blue crab in California, Hawaii, and Japan [15]. In Europe, the species was recorded for the first time in 1901 along the Atlantic coasts of France, probably introduced accidentally by ballast waters, while in the Mediterranean Sea, it appeared in 1947 in the Aegean Sea [3,15]. In the following years, the blue crab progressively extended its invasion range, with a remarkable expansion observed in the last decade in the Mediterranean and Black Seas as well as along the European and African Atlantic coasts [3,16].

A number of life-history and ecological traits likely contributed to the invasion success of *C. sapidus* [15,17,18], yet a crucial role might be attributed to its high fecundity—females produce 2 to 8 million eggs per spawning event and brood repeatedly during the year depending on water temperature [19,20,21] and high dispersal capability of larval stages in coastal environments [22,23]. In general, the high environmental connectivity of marine habitats represents one of the main factors hampering the actions generally put in place to control bioinvaders, such as those aiming at their complete eradication [24]. Functional eradication, i.e., targeting a population to decrease its density to levels determining acceptable impacts on the invaded system, is presently acknowledged as a more appropriate mitigation strategy [25,26], in particular when the bioinvaders are species of economic interest [27,28]. The opportunity to develop an economically valuable fishery while controlling the ecological impact of the blue crab has long been emphasized [27]. Recently, the General Fisheries Commission for the Mediterranean (GFCM) identified adaptation strategies to cope with the potential effects of invasive species on fisheries as target outputs of the mid-term strategy (2017–2020) [29]. Noticeably, among the strategies developed, the Recommendation GFCM/42/2018/7 established a regional program to fill scientific and research gaps on *C. sapidus* and the Lessepsian *Portunus segnis* ([30]; see also https://www.fao.org/gfcm/researchprogramme-bluecrabs (accessed on 22 September 2024)). The underlying assumption of the research program is that the integration of fishery management actions within a context of functional eradication and impact mitigation requires an advanced, context-specific understanding of the ecology and biology of the species across invaded Mediterranean countries.

A diverse set of impact-related ecological information is starting to become available at a whole Mediterranean scale on the blue crab as related to e.g., its distribution and invasion dynamics [3,16,18], multi-specific effects on recipient communities [31], trophic habits [32,33,34,35,36], contamination by pollutants [37,38,39,40,41,42], and pathogens [43,44,45]. In contrast, studies focused on the fishery biology of the species have been performed prevalently in the eastern sectors of the Mediterranean basin [46,47]. Recent investigations from Greece [47,48,49], Southern Italy [50], and Montenegro [51] are valuable exceptions, yet additional research efforts are urgently needed on the biological characteristics of blue crab populations to allow for an effective management of their Mediterranean stocks.

Accordingly, our primary aim was to contribute to fill this knowledge gap and perform a comprehensive investigation of the abundance, size structure, sex ratio, reproductive biology, and growth dynamics of the species in the Lesina Lagoon, an Italian brackish system located on the southwestern coasts of the Adriatic Sea. The blue crab was first recorded in the lagoon in 2007 and its occurrence has been repeatedly confirmed in the following years [21,52] and the species currently represents a predominant component of the by-catch of local artisanal fisheries, generally targeting finfish species such as the European eel *Anguilla anguilla*, the European sea bass *Dicentrarchus labrax*, and the big-scale sand smelt *Atherina boyeri* [53]. Blue crabs were captured using fyke nets on a monthly frequency over one year; a suite of analytical approaches and tools based on size–frequency data were used to provide an in-depth scrutiny of crucial parameters characterizing the fishery biology of the blue crab population in the Lesina Lagoon, including natural and fishing mortality rates as well as exploitation rates.

## 2. Materials and Methods

### 2.1. Study Area

The Lesina Lagoon (41.88° N, 15.45° E) is a shallow (0.7 m mean depth, 1.6 m maximum depth), micro-tidal lagoon located northward of the Gargano promontory along the southwestern Adriatic coasts of Italy (Figure 1a). The basin has a surface area of 51.4 km^2^ and is separated from the Adriatic Sea by an 18 km long sandy bar. Seawater inputs are assured by the Acquarotta and Schiapparo channels, located at the western and eastern ends of the basin, respectively, while the Lauro and Zannella streams, together with several intermittent creeks and ditches, contribute freshwater in particular along the south-eastern coast of the lagoon (Figure 1a) [21,54]. These hydrological conditions determine permanent, east-west temperature and salinity gradients [55,56]; however, a remarkable seasonal variability is generally observed in both temperature and salinity values, with minima close to 10 °C and 10 PSU observed in winter between December and February and maxima of 25–29 °C and 28 PSU occurring in summer between June and August [55,56].

Four experimental sites were identified in January 2021 in the eastern sector of the lagoon (Figure 1a), where higher blue crab abundances are generally observed as compared with the rest of the basin [57]. The sector is included in a natural reserve covering an area of 930 ha (DMAF 27/04/1981) and is a Natura 2000 site (IT9110031). The common reed *Phragmites australis* (Cav.) Trin. ex Steud dominates the riparian vegetation while the interplay between freshwater and seawater inputs affects the local salinity conditions and ultimately influences the occurrence and abundance of aquatic angiosperms [*Zostera noltei* Hornemann, 1832 and *Ruppia cirrhosa* (Petagna) Grande, 1918] and macroalgae [*Zannichellia palustris* L., *Chaetomorpha linum* (O.F.Müller) Kützing, 1845, *Myriophyllum spicatum* L., 1753, *Chara globularis* Thuiller, 1799] [58].

**Figure 1 biology-13-01051-f001:**
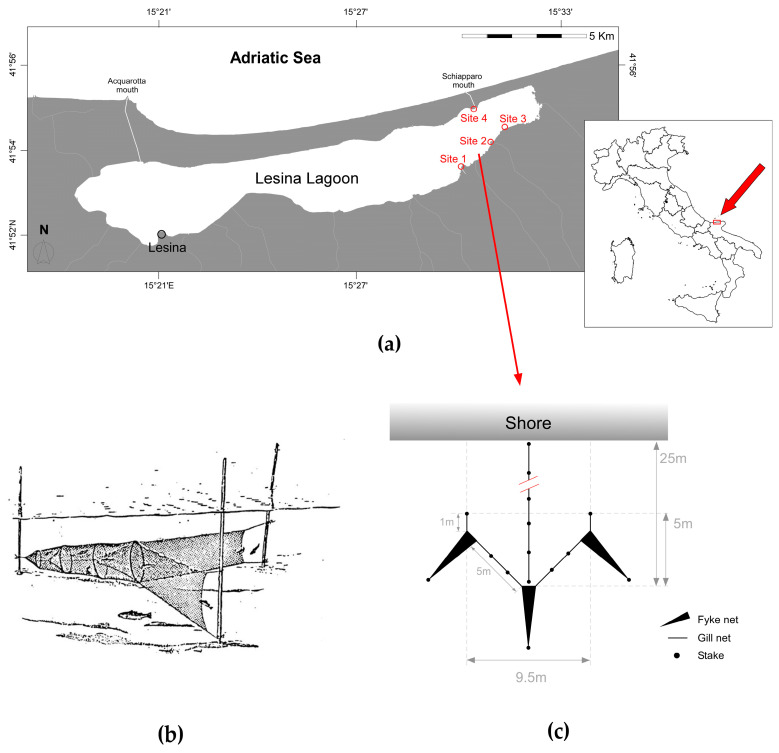
(**a**) The Lesina Lagoon. Red circles and consecutive numbers indicate the location of the four fixed installations used to capture *Callinectes sapidus* specimens during the study period. (**b**) Illustration of a typical fyke similar to those used in the study. See the text for additional details on dimensions and sampling procedures. Source: https://www.fao.org/fishery/en/geartype/226/en (accessed on 22 September 2024) [59], Food and Agriculture Organization of the United Nations. Reproduced with permission (**c**). Schematic representation of the fixed installation located at Site 1. Please note that the figure is not in scale. Identical installations were located at each of the remaining sampling sites identified in Figure 1a.

### 2.2. Sampling Procedures

Fishers operate in the Lesina Lagoon during fall and winter (September to February) using fixed gillnets, locally known as “paranze”, placed perpendicularly to the shores of the basin and conveying mobile prey into fyke nets (see Figure 1b for an example) located at regular intervals along them [60]. In this study, an experimental fixed installation based on the general operative principles of a “paranza” was assembled at each study site in February 2021. In brief, the device was made of a 6 mm mesh size, 25 m long gill net perpendicular to the shore (leader) and 3 5 m long, 1 m wide cylindrical fyke nets with an 8 mm mesh size of the final net chamber (Figure 1c). After 24 h, captured prey from each site/fyke net combination was collected and transferred to the laboratory in refrigerated containers. At each site, sampling operations were reiterated for six consecutive days; the soaking time of the fyke nets was set at 24 h, yet it varied between 24 and 72 h during the investigation due to bad weather conditions preventing field operations. In each month, on the first sampling day, a multi-parametric probe (Hydrolab DS5) was used to measure the water temperature (°C) and salinity (PSU) in triplicate close to each installation. Identical sampling procedures were repeated monthly until January 2022.

### 2.3. Laboratory Procedures

Collected brachyurans (*n* = 978) were identified to the species level; blue crabs were enumerated and sexed according to the shape of the abdominal tergites (the “apron”) [19]. The morphological maturity status of individuals was determined by inspecting the apron [61,62]: males were recognized as mature when they had an inverted T-shaped apron, while mature females had round aprons, carried eggs or has a mat of remnant brood setae. Individuals of both sexes showing triangular-shaped aprons adhering to thoracic sternites were identified as juveniles. In addition, for ovigerous females, the developmental stage of the eggs based on a three-level color scale (i.e., stage 1 = orange, stage 2 = brown, and stage 3 = black) was determined using the Virginia Marine Resources Commission color coding system (http://www.mrc.state.va.us/regulations/sponge.shtm, accessed on 22 September 2024).

Consequently, for each blue crab, the cephalothorax width (i.e., the distance between the two outermost anterolateral spines of the cephalothorax; CW hereafter) and length (i.e., the distance between the center of the anterior interorbital margin and the center of the posterior margin; CL hereafter) were measured to the nearest mm using a vernier caliper.

### 2.4. Climatic Parameters

Air temperature and precipitation data measured at 4 m above the ground level were downloaded from the website of the Agenzia Regionale per la Prevenzione e la Protezione dell’Ambiente of the Puglia Region (ARPA-Puglia; http://www.webgis.arpa.puglia.it/lizmap/index.php/view/map?repository=1&project=meteo; accessed on 11 August 2024). Data were obtained from the climatic station in the city of San Severo (FG), located approximately 20 km southward of the Lesina Lagoon (41.695006° N, 15.379159° E). Hourly air temperatures (in °C) and precipitation (in mm) were obtained for the period 1 February 2021–31 January 2022. For the sake of clarity, temperatures were averaged per month; precipitation data were cumulated over the same temporal resolution.

### 2.5. Data Analysis

When not stated otherwise, values in the text are expressed as mean ± 1SE. All data treatments and statistical procedures were implemented in the R package v. 4.4.2 [63]. For univariate parametric analyses, data normality and homoscedasticity were preliminarily checked using Shapiro–Wilks and Levene tests, respectively; when necessary, data were log- or square root-transformed to meet the assumptions.

Parameters repeatedly determined during the investigation (e.g., water parameters or CPUEs) were treated as non-independent, time-correlated variables; accordingly, differences between sampling months were tested using paired *t*-tests. Differences in ratios and deviations from 1:1 were tested for significance using χ^2^ tests. Statistical significance was evaluated at α = 0.05; when tests were reiterated, α values were adjusted by performing sequential Benjamini–Hochberg corrections for multiple tests to reduce the risk of a type-I error [64].

#### 2.5.1. Abundance

The overall abundance of blue crabs recorded during the investigation was estimated by cumulating the number of captured individuals for each site over the different sampling times on a monthly basis. Data were ultimately normalized by the total number of fyke nets per site (3) and total soak time (in h) and expressed in Catch Per Unit Effort (CPUE hereafter) as N. of individuals fyke net^−1^ d^−1^.

In June and July, when CPUE values reached maximum values (see Section 3), the spatial abundance of blue crabs was estimated for each of the four sampling sites using the Carle–Strub depletion method [65]. The method overcomes the limitations characterizing other approaches such as the Zippin removal method [66] when captures from passes two or three are higher than the first pass. The Carle–Strub method was implemented using the FSA R package [67] on four consecutive catch data with soak time = 24 h, and normalized by the expected area of action of each fixed installation (i.e., 20 m × 9.5 m = 190 m^2^; Figure 1c).

#### 2.5.2. Size Cohorts

The occurrence of distinct cohorts in the size distribution of the blue crabs collected over the whole investigation was verified using the Bhattacharya’s method [68]. The procedure was implemented in the *TropFishR* R package [69,70] on CW data by fitting Gaussian components to mode means, and decomposing the size–frequency distribution into a series of normal curves representing the cohorts. Modes were accepted as distinct cohorts if their separation index was above a critical value of 2.0 [71].

#### 2.5.3. Growth Dynamics and Mortality

In temperate regions, seasonality significantly affects the growth of decapod crustaceans, including the blue crab [72,73]. Here we used Somers’ seasonalized version of the von Bertalanffy growth function (sVBGF hereafter) to calculate the growth parameters of the crab in the study area [74,75]:$${CW}_{t}=CW\infty \left[1-e^{\left[-K\left(t-t_{0}\right)+\left(C\frac{K}{2\pi }\right)sin2\pi \left(t-t_{s}\right)-\left(C\frac{K}{2\pi }\right)sin2\pi \left(t_{0}-t_{s}\right)\right]}\right]$$
where CW_t_ is the carapace width at age t (in mm), CW∞ is the asymptotic carapace width (in mm), K is the annual growth coefficient (yr^−1^), t_0_ is the theoretical time of year when CW = 0 and spawning occurs [76], t_s_ is the time of year when the growth rate is highest, and C is the magnitude of the seasonal variation in growth, varying between 0 (no seasonal variation) and 1 (maximum variation).

Growth parameters were estimated using the *TropFishR* R package [69,70]. Specifically, CW frequency data were aggregated at a monthly level, binned into 5 mm size class intervals, and restructured after a square root transformation to identify unobservable frequency peaks [77] adopting a moving average span M of 9, 7, and 5 for the total number of individuals, and for females and males separately. M was determined considering the number of bins spanning the width of the smallest-sized cohort observed during the investigation (see Section 3) [78]. The Electronic LEngth Frequency ANalysis (ELEFAN) procedure was implemented using an advanced optimization approach based on a genetic algorithm and 1000 resamples [78]; the procedure has already been proven reliable for other brachyuran species [79,80]. The fit of the models was assessed by calculating the fitting score index Rn as follows:$$Rn=10^{ESP/ASP}/10$$
where ESP is the estimated sum of peaks, computed by summing the scored bins intersected by the sVBGF curve, and ASP is the available sum of peaks, computed as the maximum sum of positive scored bins a single curve could obtain [77]. The procedure allowed the estimation of the parameters of the sVBGF function (i.e., CW∞, K, t_0_, t_s_, and C), the maximum age, and the growth performance index (Ф’) computed as follows:$$\Phi =\mathrm{Log}\left(K\right)+2\times Log\left(CW\infty \right)$$

We estimated the total instantaneous mortality rate Z adopting the length-converted catch curve analysis and the graphical approach proposed by Pauly [81] using the CW frequency data. To calculate an approximation of the natural mortality M, due to e.g., predation or senescence, we used the sVBGF growth parameters CW∞ and K [82], while the instantaneous fishing mortality rate, F, was estimated as F = Z − M. The exploitation rate E was calculated as E = F/Z [83].

#### 2.5.4. Size at Maturity

To determine the carapace width at 50% morphometric maturity (L50) in male and female crabs, we compared two different procedures. First we directly fitted a binomial logistic regression model to CW measurements and maturity stage as classified by the inspection of the apron using the R package *AquaticLifeHystory* [84]. The second procedure was indirect and assumed the occurrence of statistically significant breaks in allometric relationships related with morphometric maturity. In the blue crabs, the morphological maturity is known to be associated with variations in the shape of the carapace in terms of length/width ratio [85,86]. Here we used the R package *sizeMat* [87] to classify the individuals in two groups (immature and mature) using a Principal Components Analysis with two allometric variables (CW = independent variable, CL = dependent variable) in log base. Individuals were assigned to each group using Ward’s hierarchical clustering procedure applied to Euclidean distances [88]. Subsequently, a linear discriminant analysis was performed to classify individuals based on their CW and CL values. The L50 was ultimately estimated using a frequentist logit procedure. Segmented regression was applied to identify breakpoints and confidence intervals at which biometric data indicated morphometric maturity in relation to the allometric relationships [87].

## 3. Results

### 3.1. Water Temperature and Salinity

Water temperatures in the study area varied three-fold during the investigation, ranging between 10.4 ± 0.1 °C (January 2022) and 30 ± 0.4 °C (July 2021), and generally mirroring the atmospheric seasonality (Figure 2a,b; Pearson R^2^ = 0.96, *p* < 0.0001, 10 d.f.). In particular, during winter months (December, January, and February), similar temperatures lower than 11.1 ± 01 °C were recorded (max *t* = 1.86, *p* = 0.16, 3 d.f. for the comparison December 2021 vs. January 2022). In summer, temperatures measured in June (26.7 ± 0.3 °C) were significantly lower than in July and August 2021 (30 ± 0.4 °C and 29.6 ± 0.5 °C, respectively; min *t* = 9.24, *p* = 0.003, 3 d.f. for the comparison June vs. July) with the latter two months showing no significant differences (*t* = 1.27, *p* = 0.29, 3 d.f.).

In contrast, considerable temperature shifts occurred in spring when values doubled from 11.1 ± 02 to 24.3 ± 0.3 °C between March and May (min *t* = 9.01, *p* = 0.003, 3 d.f. for the comparison March vs. April). Similarly, in autumn, temperatures dropped significantly from 21 ± 0.1 °C to 13.8 ± 0.1 °C between September and November (min *t* = 4.65, *p* = 0.02, 3 d.f. for the comparison October vs. November).

Water salinities ranged between 16.1 ± 1.2 PSU (April) and 29.6 ± 1.5 PSU (August) (Figure 2a). A nearly significant co-variation was observed with temperature (Pearson *r* = 0.56, *p* = 0.06, 10 d.f.); however, a less-pronounced seasonality was observed. In August, salinities were significantly higher than those observed in all the remaining months (min *t* = 4.19, *p* = 0.02, 3 d.f. for the comparison with November) with the exception of July (24.6 ± 2.1 PSU; *t* = 2.67, *p* = 0.07, 3 d.f.). In April, the lowest salinities were measured, close to those determined in May (17.1 ± 1.4 PSU; *t* = 1.41, *p* = 0.25, 3 d.f.). June, July, and November showed similar values (max *t* = 2.08, *p* = 0.13, 3 d.f., for June vs. November) ranging between 23.5 ± 1.7 PSU (June) and 25 ± 1.4 PSU (November), significantly higher than the values determined in September, October, and December, when negligible differences were observed (max *t* = 3.02, *p* = 0.06, 3 d.f. September vs. October). February, March, and January showed similar values (max *t* = 1.38, *p* = 0.26, 3 d.f.), ranging between 18.9 ± 0.7 PSU (March) and 19.7 ± 0.1 PSU (January). In winter, significant differences were observed only between December 2021 and January 2022 or February 2021 (min *t* = 5.96, *p* = 0.03, 3 d.f. for the comparison October vs. November).

### 3.2. Abundance Patterns

Overall, 838 blue crabs were collected during the investigation (Table 1). Mean total CPUE estimations ranged between 0 in December 2021 and January 2022 when no specimens were captured, and 1.76 ± 0.37 (±1SE) crabs fyke nets^−^^1^ d^−^^1^ in July.

Overall, total catches followed a seasonal pattern (Figure 3), which were significantly correlated with water temperature (Pearson R^2^ = 0.57, *p* = 0.01, 10 d.f.), but not with salinity (R^2^ = 0.34, *p* = 0.28). CPUEs progressively increased from February (0.15 ± 0.02 crabs fyke nets^−1^ d^−1^) to maximum, with similar values in June (1.51 ± 0.69 crabs fyke nets^−1^ d^−1^) and July (1.76 ± 0.37 crabs fyke nets^−1^ d^−1^; *t* = 0.31, *p* = 0.78, 3 d.f.). May showed low, apparently off-pattern CPUEs compared with those determined in April and June (0.3 ± 0.2 vs. 1.3 ± 0.4 and 1.5 ± 0.7 crabs fyke nets^−1^ d^−1^, respectively; Figure 3); however, no statistically significant differences were observed (max *t* = 2.71, *p* = 0.07, 3 d.f., for the comparison May vs. June). Subsequently, a progressive decrease occurred from August (1.18 ± 0.1 crabs fyke nets^−1^ d^−1^) to November (0.1 ± 0.01 crabs fyke nets^−1^ d^−1^), when CPUE values showed values similar to February (*t* = 0.57, *p* = 0.61, 3 d.f.). As observed for total catches, CPUEs of females and males showed a remarkable seasonal pattern of variation (Figure 3) and a significant co-variation with water temperatures was observed (Pearson R^2^ = 0.66, *p* = 0.002 and R^2^ = 0.34, *p* = 0.04 for males and females, respectively; 10 d.f.). Noticeably, while the variation of females’ abundance resulted completely independent from salinity (R^2^ = 0.003, *p* = 0.85), males showed a nearly significant relationship (R^2^ = 0.31, *p* = 0.06).

An overall spatial abundance of 0.21 blue crabs per square meter (±0.07 SE, *n* = 8) was estimated between June and July using the Carle–Strub depletion method on consecutive catch data (Appendix A, Table A1).

Estimations varied ten-fold across sampling sites and months between 0.06 and 0.64 crabs m^−2^ (site 1 and 2 in June, respectively; Table 2), yet no significant effect of the factor “month” was observed (t = 0.56, *p* = 0.61, 3 d.f.).

### 3.3. Sex Ratio and Ovigerous Females

Of the 838 blue crabs captured during the investigation, 411 were males and 427 females (Table 1), with a total sex ratio of 0.96:1 (♂:♀). The sex ratio remained close to 1 in February, March, June, July, October, and November (max χ^2^ = 2.98, *p* = 0.08; Figure 4a), while males dominated only in August (♂:♀ = 3.05; χ^2^ = 13.63, *p* = 0.0002) and September (♂:♀ = 2.55; χ^2^ = 4.51, *p* = 0.03). Conversely, females outnumbered males in April (♂:♀ = 0.61; χ^2^ = 6.52, *p* = 0.01) and May (♂:♀ = 0.41; χ^2^ = 4.86, *p* = 0.03).

Ovigerous females (N = 45) were captured for the first time in April (N = 13), when a prevalence of individuals at egg stage 1 (N = 12) and only one female at egg stage 3 were collected (Figure 4b). The highest number of ovigerous females were captured in May (N = 16) when females at egg stage 2 and 3 dominated over those at egg stage 1 (N = 5 and 9 vs. N = 2, respectively). The number of ovigerous females progressively decreased in the following summer months until August, with an increase in the dominance of individuals at stage 3; no ovigerous females were collected in the autumn and winter months (Figure 4b).

### 3.4. Size Structure

Overall, blue crabs captured during the investigation showed a mean CW (±1SE) of 146.43 ± 1.39 mm, with values ranging between 14 mm and 245 mm (Table 1). The mean CW of females (150.93 ± 30.6 mm; Table 1) was significantly larger than that of males (142.86 ± 45.11 mm; t-test with separate variance estimates: t = 3.07, *p* = 0.002, 747.77 d.f.). Conversely, males showed a higher variability in size, with values varying between 14 and 245 mm (Δ = 231 mm) while females ranged between 20 and 230 mm (Δ = 210 mm; Table 1). The analysis of the CW–frequency distribution (Figure 5) of the total number of captured blue crabs highlighted the occurrence of three main modes at CW values corresponding to 25 ± 0.67 mm (Age 0), 95.62 ± 1.45 mm (Age 1), and 171.46 ± 1.61 mm (Age 2+) (mean ± 1SD). Small-sized individuals represented by the first mode showed a non-seasonal pattern, as they occurred mainly in February and in November but were also captured in May and June (Figure 6). Medium-sized individuals included in the second mode occurred mainly in summer between June and July, while large-sized crabs belonging to the third mode were captured in spring with a peak in April and throughout the summer and autumn months (Figure 6).

The first mode was not recognized in females, given the exiguous number of small-sized individuals with CW < 50 mm (N = 5) captured only in February and June (Figure 7); thus, only two modes at CW = 111.37 ± 1.95 mm and 169.06 ± 1.11 mm were identified. Conversely, 14 small-sized males with CW values < 50 mm were collected mainly in February and episodically in May, June, September and November (Figure 7); accordingly, their frequency distribution was characterized by three modes at 2.01 ± 0.49 mm, 104.94 ± 2.81 mm, and 179.97 ± 1.75 mm. No remarkable sex-related differences in the monthly frequency distributions were evident for medium- and large-sized crabs belonging to the second and third modal classes (Figure 7).

### 3.5. Morphological Maturity

On the basis of the visual classification performed on the shape of the apron, 92 females and 122 males were identified as immature, representing 21.54% and 29.68% of the total number of captured individuals as per respective sex. Immature and mature females were characterized by a mean CW (±1SE) of 99.03 ± 2.66 mm and 164.79 ± 17.86 mm, respectively, while the mean CW of immature and mature males were 81.89 ± 2.06 mm and 167.48 ± 1.45 mm. In February, immatures represented almost the totality of the catches independently from the sex; subsequently, the number of immature individuals peaked in June and July in both sexes (Table 3); in June, in particular, they represented 63.16% of the total number of captured females and 87.38% of males. Contributions > 50% were also observed in August for females and in November for males (Table 3).

The mean carapace width at morphological maturity (L50_Apron_) of female and male blue crabs was estimated using a bootstrapped binomial model at 122.1 mm and 110.6 mm, respectively (Appendix B, Figure A1; see also Table 4 for SE and 95% bootstrapped CI).

Similar mean CL values were observed for females and males (Table 1; *t*-test with separate variance estimates, t = 0.09, *p* = 0.92, 703.07 d.f.). The classification procedure performed on CW-CL data successfully identified in both female and male blue crabs significant discontinuities that allowed an effective discrimination of two groups of individuals (Figure 7 and Figure 8). L50 values calculated on the basis of the results of the analysis (L50_CW-CL_) were fully consistent with L50_Apron_ estimations, resulting in 123.1 mm for females and in 112.3 mm for males (Figure 7 and Figure 8, fully overlapping in terms of 95% bootstrapped confidence intervals (Table 4).

### 3.6. Growth and Mortality

Table 5 summarizes the seasonal von Bertalanffy growth parameters estimated from carapace width–frequency distribution data of female and male blue crabs (Appendix C, Figure A2).

Females showed a larger CW∞ (237.78 mm CW) than males (232.56 mm CW); similarly, the growth coefficient K was higher in females (0.63) than males (0.36). A seasonal growth oscillation was more evident in females than in males (C = 0.62 vs. 0.57). The start of the slowest growth period was estimated in January for males (WP = 1). The growth performance index Ф’ was higher for females than for males (Ф’ = 2.55 vs. 2.31). The overall potential lifespan of the population was assessed in 4.8 years (Table 2), yet a remarkable sex-related difference was observed; as for females, a maximum age of 5 years was estimated, as compared to 8 years for males.

A total instantaneous mortality rate Z of 2.77 y^−1^ was estimated for the whole population, yet an almost three-fold difference was observed between sexes, with females showing the highest Z as compared with males (Table 6). Similarly, natural mortality rates M confirmed a sex-related differentiation; however, the largest dissimilarity occurred for the fishing mortality F, six times higher in females than in males (Table 6). Exploitation rates were estimated as 0.57 for the whole population, with females twice as higher than in males (0.7 vs. 0.32; Table 6).

## 4. Discussion

### 4.1. Abundance, Sex Ratio, and Size Structure

The biology of *Callinectes sapidus* has long been investigated in its native range and a seasonally-dependent pattern has been generally identified in the timing of key events, even though considerable latitudinal variations may occur [89,90]. Temperature is a critical factor influencing the phenology of the crab throughout its life cycle [89,91,92]. In addition, to complete its life cycle, the species requires both marine and brackish environments, with habitat use significantly affected by sex and ontogeny [19,93]. The results of our analyses on blue crab catches in the Lesina Lagoon are coherent with this framework and reflect the interplay of seasonal variations in water temperature with the crab’s behavior and demography [93]. Indeed, the significant relationship observed between CPUE values and water temperature, the latter in turn influenced by local climate, confirms the results of a number of investigations ultimately identifying, in seasonally-related variations in climatic conditions, a driver of blue crab abundance fluctuations [6,47,94]. More specifically, in late spring (i.e., April and May) we observed a prevalence of large-sized adult females with a CW comprised between 150 and 200 mm, likely belonging to the third modal class presumably including individuals of age 2+ (Figure 6b and Figure 7b). Females were predominant, and the majority of them were in an ovigerous condition (Figure 5a,b). In the following months until September, peak abundances of females were observed, with ovigerous females progressively shifting to different egg stages, decreasing in abundance to the point that the sex ratio ♂:♀ reached values between 2.5 and 3. During this period together with females of age 2+, specimens from the second modal class of age 1+ (CW ranging between 80 and 130 mm) constituted a significant aliquot of the captures. In temperate environments, while male blue crabs generally reside in estuaries and other brackish waters independently from their ontogenetic stage, in late spring adult females migrate after mating with neighboring marine waters to spawn [20,93]. Once released, planktonic larvae (zoea stage) are dispersed offshore; subsequently, at the megalopa stage they return to brackish systems where they settle as juveniles and complete their life cycle [20,93]. In addition, females store sperm from a single mating event and may have a lifetime reproductive potential of at least 8 broods distributed over more than 2 years [95,96,97]. Our spring and summer observations corroborate the migration of two age classes of females inseminated in preceding years and ready to leave the lagoon, as evidenced by their presence at the Schiapparo mouth. In June and July, when the maximum number of blue crabs was captured, spatial abundances were determined using the Carle–Strub depletion method on consecutive catch data. A number of efforts have been made to provide quantitative estimations of the spatial abundance of blue crab populations, and Taylor and Fehon [90] summarized available data for shallow-water, unvegetated, soft-bottom habitats from the Middle Atlantic, South Atlantic, and Gulf of Mexico. A considerable heterogeneity of results was observed due to differences in local conditions, in the ontogenetic stage targeted in the investigations, and in sampling methodologies. However, summer densities ranged between 0.1 and 0.5 crabs m^−2^ on average, with maxima between 0.12 and 2 crabs m^−2^. Our estimations were consistent with these data, as an overall spatial abundance of 0.21 crabs m^−2^ was determined, even though estimations varied ten-fold across sampling sites and months between 0.06 and 0.64 crabs m^−2^.

In the following autumn months, a decrease in catches mirrored the steep decrease in water temperature, which was likely to be determined by the departure of mated females but also by the movement of juveniles and mature males shifting to deeper waters to overwinter [6,93]. Indeed, the average water depth of the Lesina Lagoon is 0.7 m yet the eastern sector is generally shallower, with maximum water depths of 1.6 m occurring in the central and western sectors [98]. Catches in subsequent months reduced to zero when water temperatures fell under 11 °C (Figure 2). Winter temperatures exert significant control over blue crab abundance and overall population dynamics of blue crabs [91,99], for example, moulting, and thus growth ceases when the temperature falls below a minimum temperature threshold of 9–11 °C [100,101]. Accordingly, at temperate mid-latitudes where winter temperatures regularly fall below this threshold, blue crabs overwinter burrowing in the sediment in a state of dormancy, when an increase in mortality rates can be observed [99,102,103,104]. The immediate consequence is that no crabs are captured using fishing gears whose effectiveness relies on active movement of the prey, as this is the case here. Noticeably, while dormancy in winter months appears to be the norm in temperate coastal systems in the United States such as the Chesapeake Bay, in invaded Mediterranean systems, blue crabs have been shown to remain active, even though at low abundances, when water temperatures fall below 9–11 °C [47]. Accordingly, here we observed in February 2021 at water temperatures close to 11 °C (Figure 2) low abundances of juveniles belonging to the age 0 class and large-sized individuals from the age 2+ class (Figure 3 and Figure 7).

The results of our analyses provided a clear and consistent picture of the seasonal dynamics of blue crabs in the Lesina Lagoon, yet some concluding methodological remarks are necessary. Here we used fyke nets adopting a fixed-site design, generally acknowledged to provide reliable results for population-scale studies of brachyurans [89]. In addition, compared with other fishing gears (e.g., baited traps), fyke nets are considered non-selective as they capture any specimen moving along the shoreline and it is intercepted by the leader regardless of its size and sex [105]. It is evident that the differing movement behaviors between migrating females and less active males may have determined an increase in catchability of the former, at least during spring and summer months. Further, lagoon-wide investigations based on non-selective dredge or trawl surveys and adopting a stratified random design are necessary to corroborate our estimations of blue crab abundances, size structure, and sex ratio. Specifically:(1)Here we had no information regarding the catchability of blue crabs by fyke nets, thus our estimations of abundances, both in terms of CPUE and spatial density by the Carle–Strub method, might be underestimated. The catchability of a target species by a given type of fishing gear depends critically on the absolute spatial density of the species itself [106]. Here the natural density of blue crabs was unknown, and we cautiously assumed it proportional to the catches;(2)Overall, the CW frequency distribution of crabs had a tri-modal shape with three age classes assigned to age 0 (young-of-year crabs hatched during the preceding summer), age 1 (crabs one year old) and age 2+ (crabs two or more years old; Figure 6a). Blue crab populations consist of two three-year classes and possibly six to eight [61,107,108,109] as confirmed by the results of our seasonal von Bertalanffy analysis (Appendix C, Figure A2a). However, contrary to males, we did not identify for females the age 0 class due to the paucity of sampled individuals. Since the sex ratio is presumed to be balanced for blue crabs in the juvenile stage [110,111], sex-related differences in movement behavior might have influenced the catchability of fyke nets, ultimately biasing our size structure assessments;(3)We observed a sex ratio close to 1:1, with females dominating in April, May, and July (Figure 5a). Information from system-wide dredge surveys in the US performed in winter generally indicated balanced sex ratios [6,110]. Yet, a dominance of males has been reported in a variety of habitats along the Atlantic coast [84,85] and in the Mediterranean Sea [47,49,112,113], with an exception represented by studies performed in Turkey, where females prevailed [46,114]. As for the proceeding point, we argue that, beside local factors that might differently affect the survival of males and females (see further in this section), migration behavior in females coupled with variations in catchability of different fishing gears may bias the estimation of sex ratios [115], representing the ultimate determinant of the disparity of results presented in the literature.

### 4.2. Size at Morphological Maturity

The assessment of the size at which organisms reach sexual maturity represents a key step in fisheries management, as it sets the minimum size for individuals to breed at least once before being harvested, ultimately protecting the reproductive potential of exploited populations [116,117]. In addition, in invasive species management, it may provide useful information for the implementation of effective actions for functional eradication and the identification of harvest-driven trait changes in the bioinvader of interest [118,119].

Here, the results of the analyses based on the visual inspection of the apron shape indicated for females a larger L50_Apron_ than males (122.1 mm vs. 110.6 mm). The relatively scant body of information available for the blue crab confirms the occurrence of dissimilarities between sexes consistent with our findings, even though with varying degree of differentiation [50,120,121]. A number of factors might be at the origin of the observed differences, including sex-specific strategies of energy allocation to somatic and reproductive tissues during growth [122], as well as a differential sensitivity to variations in local biotic and abiotic factors such as e.g., temperature, salinity or quality and availability of trophic resources [96,123]. Local environmental conditions may also be determinants to the huge variation in size at morphological maturity observed across locations in both sexes [46,47,50]; ultimately, fishing and overexploitation may further affect the size at maturity of females, as evidenced in native US systems [124].

Noticeably, we showed that L50_Apron_ values matched those calculated using the classification procedure based on the detection of discontinuities in CW-CL data (L50_CW-CL_: 123.1 mm and 112.3 mm for females and males, respectively). In crustaceans, growth is essentially continuous, while the increase in external dimensions proceeds by a discontinuous series of moults mirrored by sudden changes in the relative growth rates of body parts [122]. The strongest variations usually coincide with morphometric sexual maturity when a metabolic and physiological trade-off between investment in somatic or reproductive tissues related with gonadal maturation occurs [125]. To our knowledge, this represents the first attempt to compare the two approaches for *Callinectes sapidus* (but see [126] for other congeners). Additionally, while several studies on *C. sapidus* as well as the congeners *C. danae* and *C. ornatus* successfully adopted a similar allometric approach comparing carapace width measurements against those of chelipeds, or the abdomen [50,126,127,128,129], no attempt has been made to date to directly use CW-CL data. This is despite the fact that Newcombe and Gray as early as 1938 [130] reported on changes in the shape of the carapace in blue crabs with the onset of maturity, and that an abrupt variation in length/width ratio characterizes both females and males after the pubertal moult [62,85,86,130]. Here we did not performed a geometric analysis of the carapace shape in captured blue crabs as those presented in other morphometric studies on *C. sapidus* and congeners [131,132]. Yet, given that the blue crab is characterized by the prominent lateral spine whose length relative to the total carapace width increases during growth [19,133], it is likely that the change in length/width ratio characterizing sexual maturity may coincide with a transition from a short- to a long-spined shape form [62].

The generality of our results must be necessarily confirmed by additional studies to verify the occurrence of morphological variability in carapace shape across different populations as that observed, e.g., for *Carcinus maenas* [134]. However, we provided a clear and statistically robust indication that allometric analysis on CW-CL data has the potential to assess morphological maturity in blue crabs with an effectiveness similar to that provided by other conventional, time-consuming procedures based on the examination of the apron or measurement of body parts.

### 4.3. Growth and Mortality

Sumer et al. [46] summarized von Bertalanffy growth parameters (i.e., CW∞, K, t0, and Ф’) for the blue crab obtained from a number of sources. These data should be taken with caution, as they were obtained on different sexes separately or on combined sexes, and, most importantly, using mainly non-seasonal von Bertalanffy models. In this study we analyzed the growth dynamics of the blue crab using a seasonal von Bertalanffy model, explicitly acknowledging the dependence of the phenology and demography of the species upon seasonal variations in local abiotic and biotic factors such as temperature and food availability [135]. Accordingly, here we discuss our results only considering studies adopting a similar methodological procedure. Of the three investigations to date available, one is a pond study performed in the USA [136] and two are on wild blue crab populations from invaded Turkish [137] and Egyptian waters [138] (Table 7). In comparison to the aforementioned references, our study showed for both sexes combined higher CW∞ and potential longevity, the lowest K, and intermediate Ф’ values (Table 6). The higher CW∞ and potential longevity, and the lower K values were confirmed for females and males separately. The comparison evidently does not allow for the identification of any general pattern, yet some issues can be addressed in terms of differences between sexes within the Lesina Lagoon.

Summer represented the period with peak growth rates for both sexes (ts = 0.51–0.52), with an evident seasonal variation of growth for females (C = 0.62) and males (C = 0.57) comprised within July and August. Noticeably, in agreement with previous studies in US waters reporting longevities varying between 3 and 8 years in relation with local environmental conditions and exploitation pressures [108,109,140,141], our estimation of the potential longevity of blue crabs was 4.8 years for both sexes combined, with females showing a lower longevity than males (5 vs. 8 years). It can be hypothesized that higher predation rates during breeding migrations, higher sensitivity to low winter temperatures, and male-induced injury during copulation [61,102] may all contribute to an increase in the natural mortality rate of females (1.16 vs. 0.87; Table 5), and ultimately a reduction in their longevity. Females also showed a fishing mortality more than six times higher than males, that can be ascribed, as previously discussed, to the interplay of female-specific behavior and fyke net catchability in the sector of the Lesina Lagoon where the present study was carried out. Accordingly, for females an exploitation rate twice as higher than that of males was determined (0.7 vs. 0.32). This depicts a condition definitely opposite to what is carried out in native areas to protect blue crab populations [142]. Thus, while the whole blue crab population in the Lesina Lagoon does not appear to be overexploited (E = 0.57), our results indicate that the fishing activities focused in the areas close to connections with the sea may actually determine a remarkable impact on females. From a conservation perspective, this represents an undeniable positive condition, as the capture of mated females might effectively reduce the reproductive potential of the population and ultimately its abundance. In the perspective of exploiting the species as a shellfish product, however, it implies an evident risk of overfishing. A thorough assessment of the advantages and disadvantages of alternative strategies of exploitation of the blue crab in the lagoon is necessary, and we strongly recommend the implementation of an integrated strategy identifying an Optimum Blue Crab Yield (OBCY), taking as a reference what has been proposed for the invasive lionfish *Pterois* spp. in the Gulf of Mexico [143]. OBCY may set a target for catches of the entire population and for females only, exceeding the conventional maximum sustainable yield (MSY) but still providing a relatively high sustainable yield. This approach contributes to population suppression and the mitigation of its impacts on the recipient system, exceeding the achievable results of targets set at or below MSY.

## 5. Conclusions

In the present study, we performed a comprehensive scrutiny of the biological characteristics and life history traits of the population of *Callinectes sapidus* in the Lesina Lagoon. In addition, we casted the results of the investigation in an integrated perspective acknowledging the need of controlling the abundance of the species to mitigate its ecological impact. We are confident that the present study may provide a useful basis for the implementation of standardized, cost-effective, Mediterranean-scale management plans focused on *C. sapidus* as well as other invasive portunid brachyurans such as *Portunus segnis* [144], matching exploitation with sustainable conservation and protection of coastal ecosystems.

## Figures and Tables

**Figure 2 biology-13-01051-f002:**
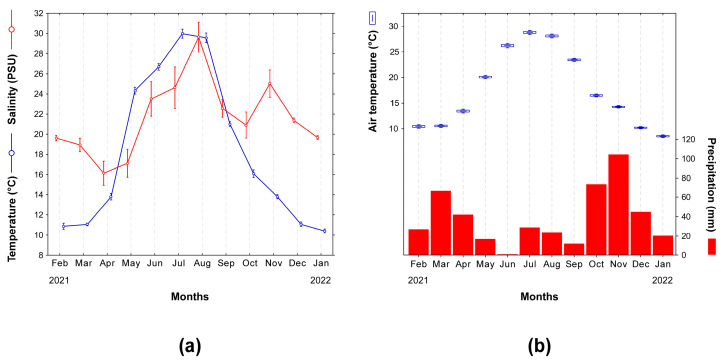
(**a**) Variation in mean (±1SE, *n* = 4) water temperature (°C) and salinity (PSU) in the study area during the investigation. (**b**) Air temperatures (°C) and precipitations (mm) measured at the ARPA-Puglia climatic station located in the city of San Severo (FG, Italy) during the study period. See the text for additional details. Air temperatures are expressed as monthly means of hourly measurements ± 1SE (boxes); whiskers are 95% confidence intervals. Precipitation data are cumulated at a monthly scale.

**Figure 3 biology-13-01051-f003:**
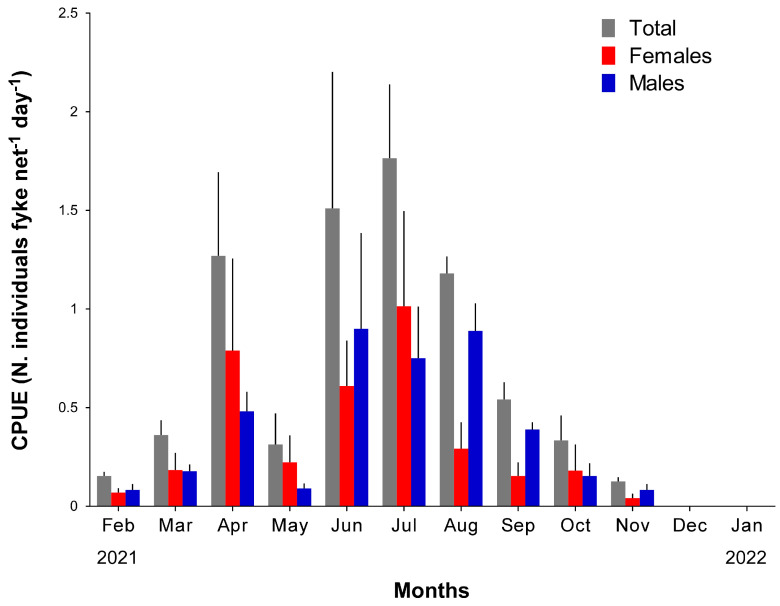
Variation in mean (±1SE, *n* = 4) monthly catches per unit effort (CPUE, measured as N. individuals fyke nets−^1^ d^−1^) of blue crabs determined during the investigation. CPUE values are reported for sexes combined and for the two sexes separately.

**Figure 4 biology-13-01051-f004:**
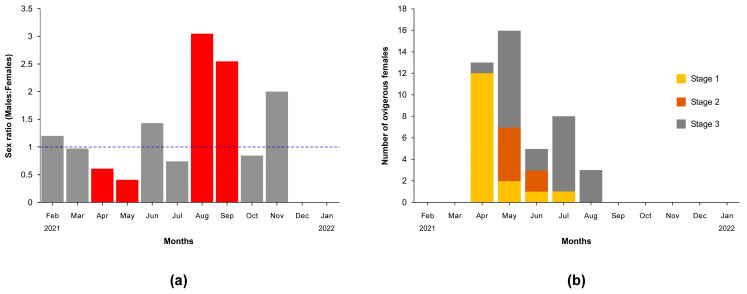
(**a**) Sex ratio of blue crabs captured during the investigation. The blue dashed line identifies the 1:1 ratio. Red bars indicate significant departures from a 1:1 ratio (χ^2^ test, *p* < 0.05 after correction for multiple tests) indicated by the blue dashed line. Statistical results for February and November 2021 should be considered with caution given the low number of total specimens collected (<10). (**b**) Number of ovigerous females at three color-determined egg stages (stage 1 = yellow; stage 2 = brown; stage 3 = black) captured during the investigation.

**Figure 5 biology-13-01051-f005:**
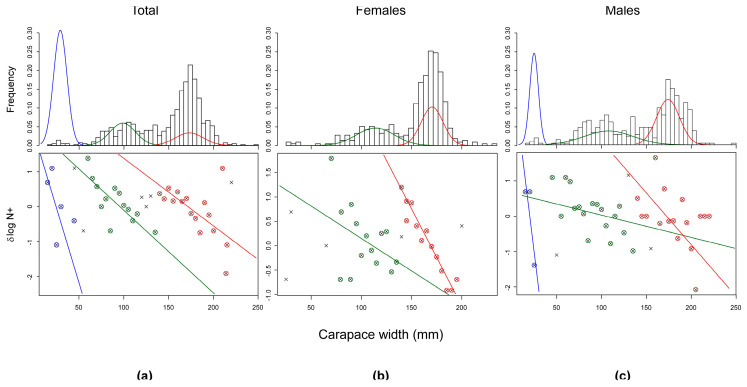
Frequency distributions of the carapace widths of blue crabs captured during the investigation for sexes combined (**a**), females only (**b**) and males only (**c**). Frequency distribution histograms include the modes identified by the modal analysis performed using the Bhattacharya method (bottom plots). See text for additional details.

**Figure 6 biology-13-01051-f006:**
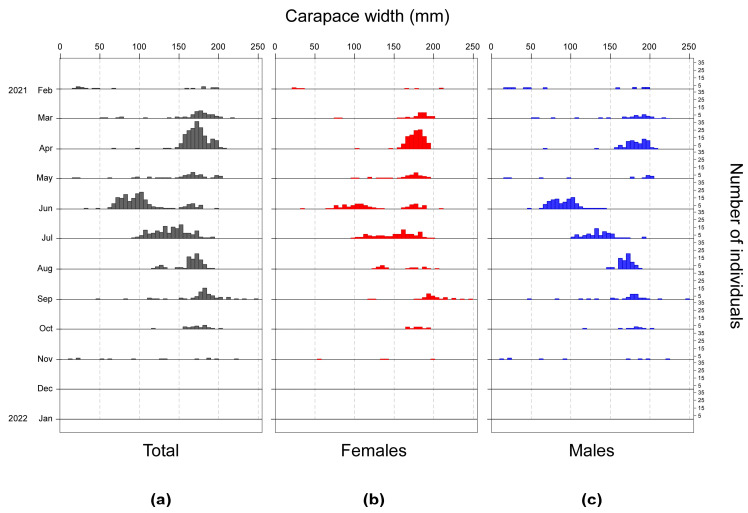
Monthly frequency distributions of the carapace widths of blue crabs captured during the investigation expressed in terms of sexes combined (**a**), females (**b**), and males (**c**).

**Figure 7 biology-13-01051-f007:**
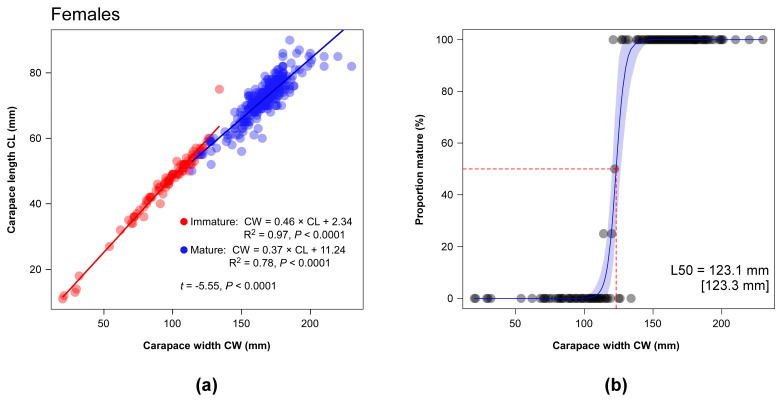
(**a**) Relationship between carapace widths and lengths (in mm) in female blue crabs; linear regression models are fitted to individual data subjected to a classification procedure (see text for details). The regression equation and the total explained variance (R^2^) are provided for each linear model; the results of an ANCOVA testing for statistically significant differences in the slopes of the two models are included. (**b**) The mean size at maturity (L50) of female blue crabs estimated using a bootstrapped binomial model applied to individuals classified as mature or immature through the analysis of carapace width–length relationships (L50_CW-CL_). Shaded areas represent 95% bootstrapped confidence intervals. L50 values estimated using a Bayesian procedure are reported in square brackets.

**Figure 8 biology-13-01051-f008:**
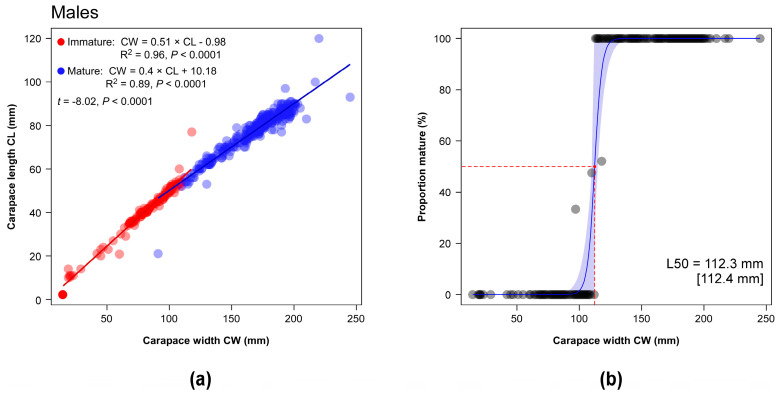
(**a**) Relationship between carapace widths and lengths (in mm) in male blue crabs; linear regression models are fitted to individual data subjected to a classification procedure (see text for details). The regression equation and the total explained variance (R^2^) are provided for each linear model; the results of an ANCOVA testing for statistically significant differences in the slopes of the two models are included. (**b**) The mean size at maturity (L50) of male blue crabs estimated using a bootstrapped binomial model applied to individuals classified as mature or immature through the analysis of carapace width–length relationships (L50_CW-CL_). Shaded areas represent 95% bootstrapped confidence intervals. L50 values estimated using a Bayesian procedure are reported in square brackets.

**Table 1 biology-13-01051-t001:** Number of blue crabs captured during the investigation. Mean carapace width and lengths are included; standard errors are in parentheses while min–max ranges are in square brackets.

	Total	Females	Males
N. individuals	838	427	411
Carapace width (mm)	146.4 (1.4) [14–245]	150.6 (1.6) [20–230]	142.1 (2.3) [14–245]
Carapace length (mm)	66.4 (0.6) [5–120]	66.3 (0.6) [11–90]	66.4 (0.9) [5–120]

**Table 2 biology-13-01051-t002:** Mean spatial abundance of blue crabs (crabs m^2^; standard error in parenthesis) captured at the four sampling sites in June and July estimated using the Carle–Strub depletion method (see text for details).

Site	June	July
1	0.64 (0.11)	0.19 (0.04)
2	0.06 (0.01)	0.08 (0.01)
3	0.14 (0.01)	0.12 (0.04)
4	0.09 (0.02)	0.25 (0.04)

**Table 3 biology-13-01051-t003:** Number of immature female and male blue crabs as identified by the shaper of the apron collected during the investigation. The contribution (in %) to the total captures as per the respective sexes are reported in parentheses.

Month	Females	Males
February 2021	4 (80)	6 (100)
March	1 (3.13)	4 (14.29)
April	1 (0.81)	1 (1.33)
May	2 (5.41)	4 (28.57)
June	48 (63.16)	90 (87.38)
July	24 (28.24)	10 (14.71)
August	11 (52.38)	---
September	---	2 (6.06)
October	---	---
November	1 (25)	5 (55.56)
December	---	---
January 2022	---	---

**Table 4 biology-13-01051-t004:** Mean carapace widths at morphological maturity of female and male blue crabs (SE in square brackets; 95% bootstrapped confidence intervals are included) estimated through the visual classification performed on the shape of their apron (L50_Apron_) and through the analysis of carapace width–length relationships (L50_CW-CL_). The total variance explained by the logistic models with a binomial error structure fitted to the data (R^2^) is reported.

	Sex	R^2^	Mean (mm)	95% CI (mm)
L50_Apron_	Females	0.82	122.1 [2.2]	115.7–125.8
	Males	0.93	110.6 [1.6]	107.2–113.9
L50_CW-CL_	Females	0.95	123.1 [1.8]	120.2–126.5
	Males	0.96	112.3 [1.7]	110–114.8

**Table 5 biology-13-01051-t005:** Seasonalized von Bertalanffy growth parameters determined for female and male blue crabs in the Lesina Lagoon. Estimations for sexes combined are included.

Sex	t_s_	CW∞ (mm)	Age Max (y)	K (y^−1^)	t_0_	C	Ф’	Rn
Females	0.51	237.8	5	0.63	0.53	0.62	4.55	0.77
Males	0.52	232.6	8	0.36	0.75	0.57	4.29	0.58
Total	0.84	236.7	4.8	0.66	0.62	0.42	4.57	0.87

**Table 6 biology-13-01051-t006:** Estimated total, natural, and fishing instantaneous mortalities (Z, M, and F, respectively, in y^−1^) for blue crabs in the Lesina Lagoon, as referred to sexes combined, and for females and males alone. Estimations of exploitation rates (E) are included.

Parameter	Total	Females	Males
Z	2.77 ± 0.33	3.88 ± 0.43	1.28 ± 0.14
M	1.19	1.16	0.87
F	1.58	2.72	0.41
E	0.57	0.7	0.32

**Table 7 biology-13-01051-t007:** Summary of the literature information on seasonalized von Bertalanffy growth parameters determined for the blue crab. Data on sexes combined and for female and male only are reported.

Sex	CW∞ (mm)	Age Max (y)	K (y^−1^)	t_0_	C	Ф’	Ref.
Total	201.6	3.7	0.81	0.22	0	4.52 *	[17]
	207.5	---	1.19/1.71	0.15–0.31	0.94/1.0	4.71–4.86	[136]
	226.4 ***	2.11	1.38	0.15	---	4.85 *	[138]
Females	206.6	4	0.74	0.24	0	4.50 *	[17]
	181.9	2.72 **	1.1	0.85	0.93	4.52	[136]
	227.6 ***	2.07	1.51	0.15	---	4.89 *	[138]
Males	209.5	6	0.5	0.36	0.2	4.34 *	[17]
	230.1	3.49 **	0.86	0.16	0.29	4.72	[136]
	215.5 ***	1.96	1.59	0.14	---	4.87 *	[138]

* CW∞ originally expressed in cm; values were re-calculated using the equation F’ = Log(K) + 2 × Log(CW∞); ** calculated as 3/K [139]; and *** calculated from CL∞ data and CW–CL relationships provided in the publication.

## Data Availability

Data are available upon request from the corresponding author; they are not publicly available due to ongoing comparative analyses.

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
