# Peer review of "Abundance, Size Structure, and Growth of the Invasive Blue Crab Callinectes sapidus in the Lesina Lagoon, Southern Adriatic Sea"

_biology, 2024, doi:10.3390/biology13121051_

Round 1

Reviewer 1 Report

Comments and Suggestions for Authors

The article includes original analysis that provides new data on the Callinectes sapidus biology in the Mediterranean Sea. These findings are significant and enrich the field of study.

I think that the authors employ appropriate statistical methods for data analysis, ensuring the validity and reliability of the results obtained. However, I am not able to rigorously evaluate all the details.

I think the writing is clear and structured. The authors provide the necessary information and the article is solid in all aspects. The article presents a thorough review of relevant studies, establishing a solid and up-to-date framework on the topic. The selection of studies is pertinent and broad. However, I suggest that authors consult and include in their manuscript the recently published article:

Vivas, M., García-Rodríguez, E., Muñoz-Vera, A. et al. Effect of the Invasive Blue Crab (Callinectes sapidus Rathbun, 1896) in a Protected Coastal Lagoon. Estuaries and Coasts 48, 9 (2025). https://doi.org/10.1007/s12237-024-01436-6

The article constitutes a valuable contribution to the scientific literature. Its comprehensive approach, methodological rigor and practical implications make it a strong candidate for publication. I recommend its acceptance, as it will offer important insights into the biology of this species so well established in the Mediterranean Sea.

Author Response

Comment 1: The article include original analysis that provides new data on the Callinectes sapidus biology in the Mediterranean Sea. These findings are significant and enrich the field of study. 

Response1: thanks a lot for the appreciation

Comment 2: I thinks that the authors employ appropriate statistical methods for data analysis, ensuring the validity and reliability of results obtained. However, I am not able to rigorously evaluate all the details.

Response 2: we made an effort to adopt the most robust and updated methodologies currently in use to investigate the fishery biology brachyurans, andto presents the results in a clear and compelling way.

Comment 3: I think the writing is clear and structured. The authors provide the necessary information and the article is solid in all aspects. The article presents a thorough review of relevant studies, establishing a solid and up-to-date framework on the topic. The selection of studies is pertinent and broad. However, I suggest that authors consult and include in their manuscript the recently published article: Vivas, M., ...et al.  https://doi.org/10.1007/s12237-024-01436-6

Response 3: accepted. The citation has been included in the revised version of the manuscript.

Comment 4: The article constitutes a valuable contribution to the scientific literature. Its comprehensive approach, methodological rigor and practical implications make it a strong candidate for publication. I recommend its acceptance, as it will offer important insights into biology of this species so well established in the Mediterranean sea.

Response 4: thanks a lot for the appreciation.

Reviewer 2 Report

Comments and Suggestions for Authors

This manuscript: "Abundance, size structure, and growth of the invasive blue crab Callinectes sapidus in the Lesina Lagoon, southern Adriatic Sea," has been well organized and written.  

However, to improve the manuscript, the following some inputs to improve:

1. In the introduction section;   If trend data on CPUE for the last 5 years is available, it should be presented to support the stock status of blue crabs in the study area.

2. Please explain in more detail what is the target species or main fishery of the artisanal fishery in the study area, that the blue crab (Callinectes sapidus) currently represents a predominant component of the by-catch (see L106)

3. In the methods section;  Please add information/explain the number of crabs samples measured in the laboratory analysis (see L164).

Reviewer 3 Report

Comments and Suggestions for Authors

The ms biology-3304045 describes about population study on Callinectes sapidus in  Lesina Lagoon.

Please make all changes in the revised ms essentially in track change mode. 

Followings are the observations

The major concern is I found a similar article on the same crab at https://www.mdpi.com/2077-1312/11/3/462 describing the similar issue in Evros River. However, it is not very clear why the study was done. The fishery biology of the crab will be different in different sectors of the same river and similar place of different rivers.

May be the growth and abundance pattern of the crab with the environmental conditions be correlated so that an optimum culture condition of the crab be standardized for their culture.

But the article is written just form population point of view. Therefore, it lost the essence of any scientific application rather than a report. As data on the population and biology of this crab are plenty, so the current article lost its true value.

General comments

Make the objective to find an optimized condition for the highest growth of the crabs in any specific season. This should be also substantiated with the existing literature. Such as any seasonal study on any euryhaline crab including Scylla, Calinectes with respect to environmental stress factors. Some of the studies done in other euryhaline crabs in lagoon are https://pubmed.ncbi.nlm.nih.gov/22299581/,  https://iopscience.iop.org/article/10.1088/1755-1315/1251/1/012048/pdf , https://pubmed.ncbi.nlm.nih.gov/23122870/, etc. Add them and make your own hypothesis as follows.

Then make the gap that such studies on the growth pattern in crabs are required in their invasive habitats and we hypothesize that the summer season would be preferred by the crabs for their better growth as the salinity of the river would be high and Callinectes prefer high salinity.

Accordingly the introduction needs to be changed. And the environmental data if not studied may be collected from other study i.e. as secondary data as they have collected form http://www.webgis.arpa.puglia.it/lizmap/index.php/view/map?repository=1&pro- 183

 ject=meteo;.

I definitely suggest including a statistics section.

The discussion is too lengthy and it is a research article. So, make it as comact as possible.

And I would like to see the conclusion as that the invasive crab can be better grown in xxx salinity, xx temperature and in y season in the Lesina Lagoon, May be in 2-3 lines.

Round 2

Reviewer 3 Report

Comments and Suggestions for Authors

The authors have cited their own 24 references and have declined most of the relevant comments that would really useful to improve the ms. Most of the replies given are out of context and the authors probably didnot understood the topic as well. Therefore, they are encouraged to understand the earlier comments, revise the ms, and add a solid hypothesis. Just one species has invaded so their biology would be studied is out of context also.  So, re-read the earlier points, and make a vigorous revision. Else the ms wont stand alone to be published. 
